# A Sustainable Risk Management Model and Instruments for Young Farmers in EU Agriculture

Aleksandra Figurek [1],*, Katerina Morphi [2] and Alkis Thrassou [1]

1   GNOSIS Mediterranean Institute for Management Science, School of Business, University of Nicosia,
    2417 Nicosia, Cyprus; thrassou.a@unic.ac.cy
2   Department of Accounting, Economics and Finance, School of Business, University of Nicosia,
    2417 Nicosia, Cyprus; morphi.k@unic.ac.cy
*   Correspondence: figurek.a@live.unic.ac.cy

**Abstract:** Young farmers are ready to embrace innovation, smart agriculture, and science-based research to ensure that their work has long-term viability, profitability, and productivity. To prevent the farming population from aging and to ensure farming success, financial support must be provided through instruments that are specifically aimed at young farmers. It is necessary that youth have access to agricultural land policy throughout the EU for the agricultural sector to remain sustainable. In the European Union, young farmers manage farms that are in the lowest-size group. The limited financial possibilities available to young people in the EU exacerbate this. The relationships between risk management instruments and other interventions (such as direct payment and ex-post instruments) and the integration of these tools into national and EU policy frameworks must be thoroughly examined in future policy. Evaluating the possible repercussions of risk management being widely implemented as well as measuring the anticipated variations in farm revenue and the volatility of agricultural commodity prices are equally crucial. Therefore, the present article utilized extant data to conduct a comparative analysis and ultimately present a set of multidisciplinary and quantitative indicators of supportive measures for young farmers in the EU, while also identifying the requisite areas for improvement.

**Keywords:** young farmers; sustainability; data; risk management; knowledge; EU

## 1. Introduction

In the European Union, approximately 11% of all farm holdings are managed by farmers under the age of 40 [1], which is down from 14%, according to earlier data from the CAP2 reform overview for 2014–2020 [2]. With fewer young people employed in agriculture, numerous factors have been proposed as the cause of the decline. Some of them have to do with young people who are prevented from entering the agricultural industry by obstacles, including high land prices, challenges in securing financing and policy support, and other issues that prevent farm succession to future generations [3].

Williamson and Petkovic claimed that uncertainty seems to exist because young farmers have to deal with several administrative issues and procedures that must be followed for them to participate in youth programs. Along with higher production costs, there is also a lack of official support in the form of capital, stability, and assurances, and finally a lack of society's acceptance of young farmers. Even with the creation of many EU projects targeted at solving the "young farmer dilemma", the issue still exists in European agriculture [4–10]. Furthermore, the data showed that, when it came to transferring farms from one generation to the next, there was no discernible replacement rate [9]. Transferring a farm from the older to the younger generation involves several procedures, including retirement, succession, and inheritance [11].

Developing replacement representations, moving up the farm ladder, and changing farm business trajectories are three interconnected processes that are crucial to comprehending succession outcomes and are taken into consideration by the stochastic succession cycle model. Successor identity is developed through one's involvement on the farm, and it is strengthened by one's connections to the farm, its location, and family background. On the other hand, progression up the farm ladder explains how young people strengthen their sense of succession by participating in more complex agricultural operations and decision-making tasks [12,13].

In academic study, farm rotation is typically regarded as a multifactorial process. Fischer and Burton (2014) identified several factors involved in it [14]. This approach has been used in research that looked at the factors that affect the chances of a farm being successfully passed down to the next generation. The factors (Figure 1) that are important to take into account when examining the probability of a farm being successfully passed down to the next generation are as follows:

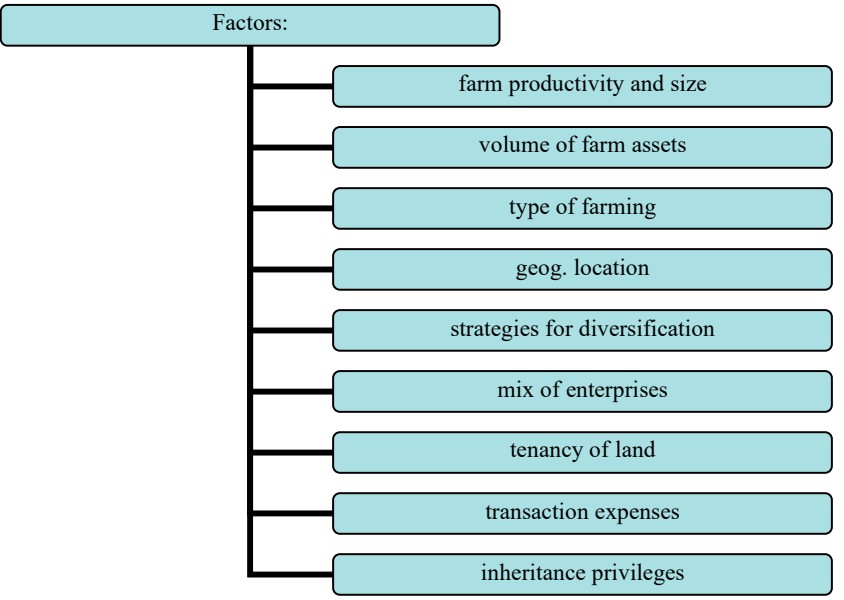

**Figure 1.** Significant elements that influence the likelihood of a farm being effectively passed down to the following generation (authors' own development).

In order to maintain output and to ensure the sustainability of agriculture and food production, young people should continue to live in rural areas and work in agriculture [15,16]. Sustainability in agricultural development is defined by the Food and Agriculture Organization (FAO) of the United Nations as the management and conservation of the foundation of natural resources as well as the direction of institutional and technological change in a way that ensures the satisfaction of human needs for present and future generations [17]. This type of development protects genetic resources for plants, animals, and the environment and is sound technically, financially, and socially.

Numerous issues, such as water pollution, land degradation, biodiversity loss, extreme weather events, and climate change, have an influence on agriculture and the food chain [18,19]. Climate change is predicted to have substantial socioeconomic effects and provide a major obstacle to attaining fair and sustainable agricultural growth. The United Nations announced the Sustainable Development Goals [20] in 2015, and these difficulties are related to rainfed agriculture.

Regardless of the differences in various definitions of agricultural vulnerability, they all include exposure, sensitivity, and adaptability as fundamental components (Figure 2). Vulnerability is defined as the degree to which a system is susceptible to changes in the climate [21]. Adjusting risk exposure, risk absorption, and the ability to bounce back from exposure-related losses all depend on one's ability to adapt. Adaptive capacity may also

be defined as the inclination or propensity to sustain harm. Therefore, to lessen their susceptibility to stress, it is imperative to strengthen young farmers' adaptive potential and minimize their sensitivity [22]. The ability to adapt differs between environments and capabilities, and it is closely associated with elements of societies, organizations, infrastructure, social, governmental, and economic factors, as well as health, education, technology, and cognitive features [23].

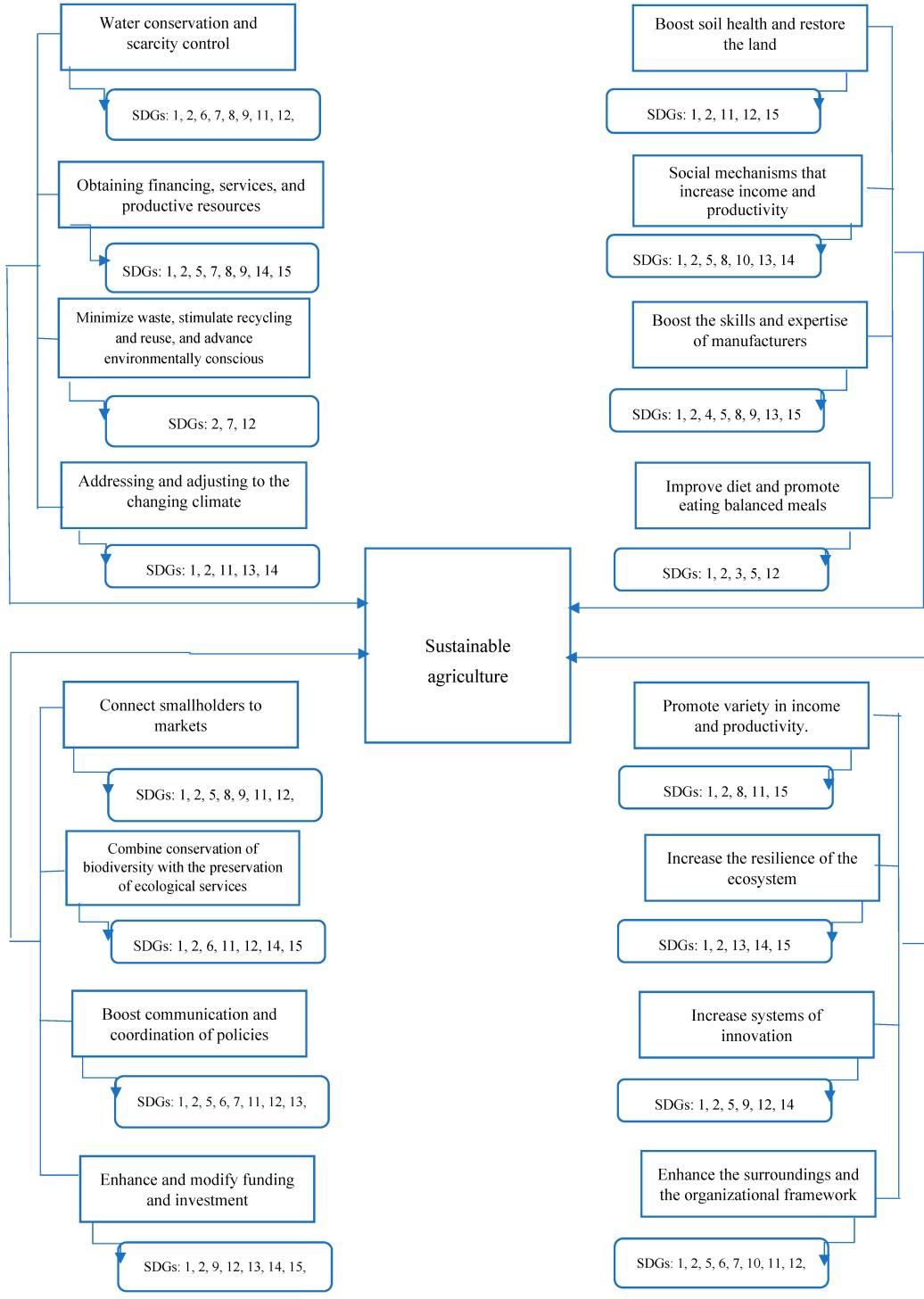

**Figure 2.** Sustainable agricultural activities (authors' own development according to FAO Sustainability Development Goals).

Farmers should have access to knowledge transfer and consulting services in areas such as technical, economic, environmental, and social understanding about succession problems. Free and public advisory services in a range of fields (including land management, inheritance law, business support, marketing counsel, and other financial guidance) may be made available to help young farmers get ready. The production risk in the EU varies substantially throughout member states and is contingent upon the size and kind of farm. Member states offer ex post help even if the current risk management strategies are insufficient to prevent a decline in revenue. Some strategies that could be used to support national systems include a unified regulatory framework, suitable control mechanisms, public reinsurance, the partial subsidization of national systems, and the assistance or subsidization of database composition at the farm level.

Providing young farmers with a solid knowledge base that includes the economic indicators needed for risk management in agriculture and support is the primary goal in ensuring the sustainability of agricultural output. In addition to substantially contributing to the theoretical delineation of this complex topic, providing a scientific foundation and direction for future research, and practically assisting the industry and policymakers in their efforts, this article identifies and presents a set of multidisciplinary and quantitative indicators of supportive measures for young farmers in the EU.

## 2. Methodology

A comprehensive analysis of the indicators and variables that impact the existence of certain constraints on the functionality of an agricultural management system necessitates a collaborative effort. When it comes to minimizing the effects of risks and limiting factors that affect productive activities within the agricultural sector, employing a systematic approach to problem solving is linked to making decisions based on good reasoning. A comprehensive approach is necessary to examine the indicators and factors that influence the existence of specific limiting features in a functioning system.

Comparative analysis was utilized to determine the elements that mitigate the risk in agriculture with the appropriate management tools and interventions, as well as to weigh the benefits and drawbacks of developing new conceptual–methodological and application solutions. Additional data processing, analysis, and presentation on youth in agriculture were based on the Eurostat database.

The Eurostat dataset [24] was specifically combined in this study to evaluate the following: (1) farm managers by gender and age class; (2) the economic size of holdings owned by young farmers; (3) agricultural labor input; (4) the economic performance of holdings owned by young farmers; and (5) agricultural output and the intermediate consumption of agricultural products. An annual work unit (AWU) is equivalent to the labor of one full-time employee on an agricultural property. The term "full-time" refers to the minimal minimum of hours required by national labor regulations that govern employment contracts. It represents the labor of a single employee who has spent the last 12 months working on the holding permanently. A worker puts in 1800 h a year at work (225 working days × 8 working hours per day). In agriculture, one way to quantify labor productivity is by using the agricultural factor of income per yearly work unit. Agricultural factor income measures the revenue generated by farming (own labor, capital, and land) to pay for both owned and borrowed production factors (such as capital, salaries, and land rentals).

The aim of the governments of the EU member states is to assist the agriculture sector in achieving shared objectives, as outlined in their strategic policies. There are large and complicated differences in the realization of agricultural income farms. These are the outcome of a number of characteristics unique to the agriculture sector, such as the economic potential, the nature of agricultural production, the expertise of agricultural producers, etc. They also serve as a representation of the outcomes of outside influences, such the socioeconomic environment.

The EU evaluates agriculture's economic status by considering the microeconomic and macroeconomic dimensions of total agricultural production. The European Union Statistical Office, or Eurostat, created a standardized process for creating specific indicators that allow EU member states to track and contrast economic statistics in this area. Priority goals are accomplished through the appropriate distribution of income among priority groups or regions and the adequate allocation of resources, the use of which is being made to achieve a higher value of agricultural production. Developing a suitable policy for the agriculture sector that will aid in its growth is a difficult and complex undertaking. Consistency in carrying out production operations associated with all different kinds characterizes agricultural production.

The European System of Integrated Economic Accounts' Economic Accounts for Agriculture (EAA) serves as the foundation for the computation of indicators of the realization of income in the agricultural sector (ESA). All forms of agricultural production and goods resulting from primary or secondary activity are included in the data that cover the agricultural sector. Non-agricultural data, such as earnings, salaries, social benefits, and asset revenue, are not included. The EAA provides an understanding of the nature of agricultural production in EU member states, the function of inputs or input parts in the development of the production process, the value of output, the cost of inputs and end products, and so on. The value of crop production, the value of cattle and livestock products produced, the value of agricultural services, and the value of integral secondary activities carried out on agricultural farms and within agricultural enterprises are all included in the valuation of agricultural production, or output. This is stated in terms of basic pricing, which does not include any product taxes and contains product subsidies.

In agricultural production, interphase consumption refers to the total amount of resources used in a given year. These resources might include energy and lubricants, fertilizers and soil development agents, animal feed, seeds and seedlings, pesticides, and plant protection agents, among others. If an asset's useful life is shorter than a year, then the tools and other fixed assets are listed here. Fixing asset acquisitions are represented as gross investments in fixed capital; thus, they are excluded from intermediate consumption calculations. Purchase prices that include taxes (minus the deductible amount of VAT) but no subsidies are used to determine the value of intermediate consumption. As a result, all goods and services used as inputs in the manufacturing process are represented by intermediate consumption, with the exception of fixed assets, which are represented by fixed asset consumption.

The formation of a sustainable risk management model should represent the basis for overcoming the risks that characterize agricultural production, as well as defining the necessary support for young farmers. The model should indicate the possibility of considering the functional elements that support the making of adequate management decisions based on knowledge, as well as application or compliance with the appropriate guidelines, which refer to farm management, as well as the support of young agricultural producers to maintain sustainable agricultural production.

## 3. Analysis and Results

### 3.1. Young Farmers in EU Member States

Employers in the EU in 2020 made up 8.7 million people, or roughly 4% of the labor force, who worked in agriculture. In Romania, where they employ somewhat more than one in five (20%) of the workforce, as well as in Bulgaria (16%), Greece (about 10%), and Poland (0.9%), agriculture is a major employer (Figure 3).

Within the 9 million holdings in the EU in 2020, men made up slightly more than two thirds (68%) of farm managers. When looking at both genders combined, the majority of farm managers (57%) had a minimum age of 55. Notably, young farmers—defined as those under 40 years old—accounted for only 11.9% of farm managers (both sexes combined).

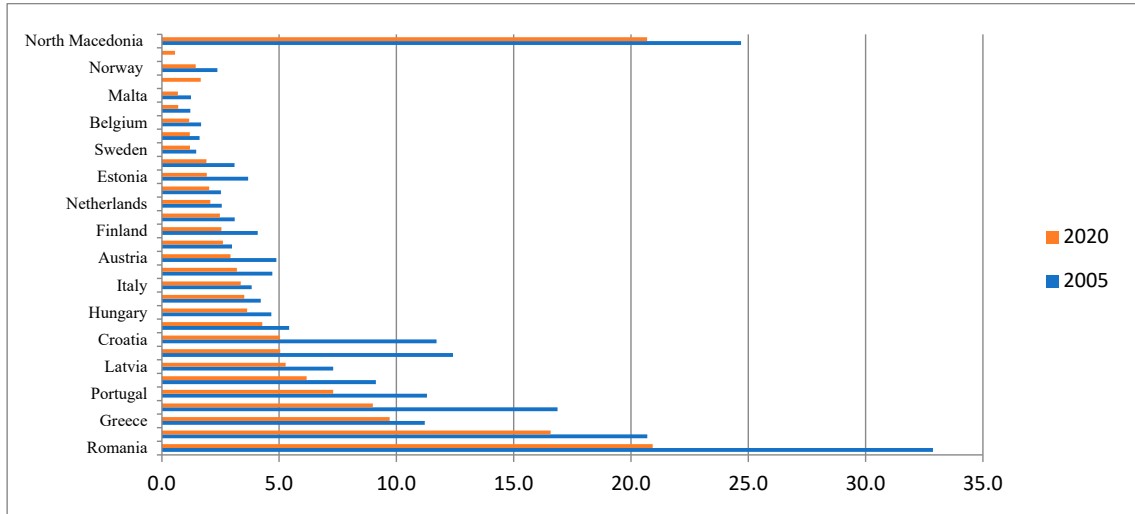

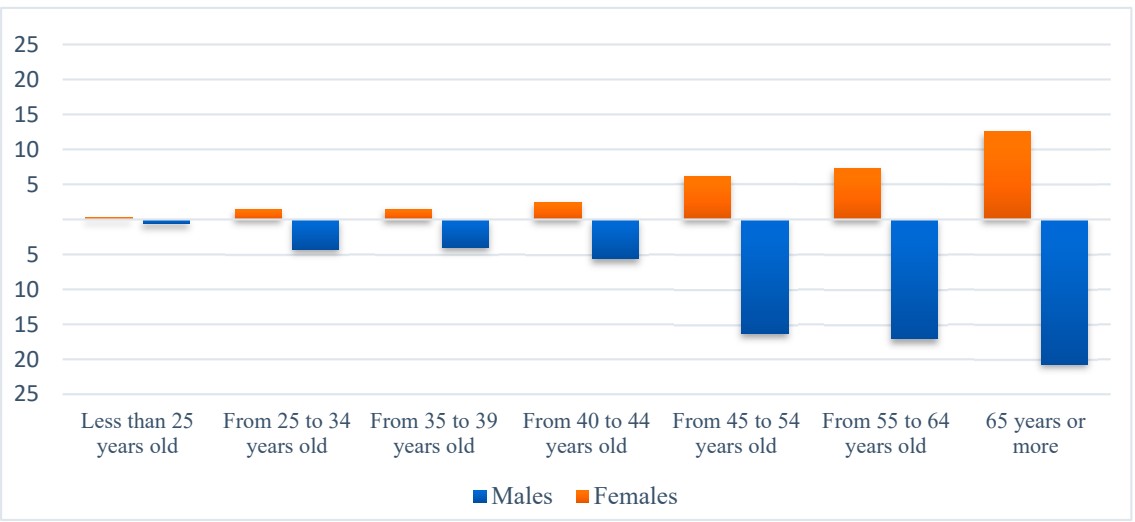

**Figure 3.** Agricultural sector employment (% of total employment, 2005 and 2020) and farm managers' ages and genders in percentages (authors' analysis based on Eurostat data [24]).

The analysis of the data showed that a higher share of young farmers (31.1%) managed medium- and large-sized farms (Figure 4). The percentage of farm managers dropped with age; around 10% of farmers 65 and older managed medium-sized and large farms. This discrepancy could be partially explained by younger farmers having higher educational attainment in terms of comprehensive agricultural training and having completed more recent professional training courses, such as those teaching cutting-edge farming techniques.

The European Commission is concerned that a lack of young farmers may harm European agriculture's competitiveness and food security in the future. In order to address the scarcity of young farmers and the elderly farming population, the European Union has developed a special initiative over the last thirty years. The European Union responded to the challenge by launching numerous projects designed to inspire youth to pursue careers in agriculture. In order to appropriately address the issue, this policy was founded on a series of legislative acts, including decisions and regulations. These include financial resources made available to young farmers by member states, such as a bonus of up to 25% on top of their CAP subsidies from the European Common Agricultural Policy (CAP) to help farmers under 40 start a business, and rural development initiatives that provide guidance and training programs.

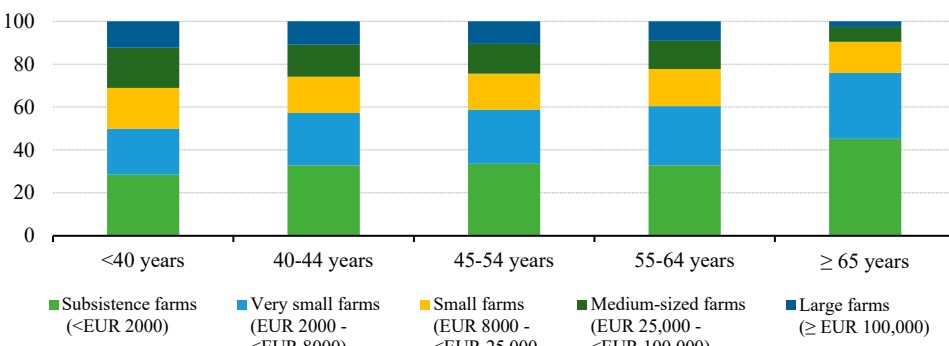

**Figure 4.** Farm managers' percentages broken down by economic size and age range (authors' analysis based on Eurostat data [24]).

Building on past attempts, the new Common Agricultural Policy uses two distinct but complementary techniques to attain this goal: (a) early retirement and (b) new entrant schemes. While the latter aims to financially support new farmers as heads of agricultural holdings, the former gives financial incentives to elderly farmers to retire early and transfer their farming operations to younger farmers. These schemes make it easier for newbies to get started as well as to change the structure of their holdings once they are up and running. When the two techniques are compared, the early retirement program appears to have a lower positive impact than new enrollment schemes [25].

With subsistence households and low levels of agricultural income, the smallest farms are typically managed by older farm managers. These farms are typically measured economically or in relation to their agricultural area. In 2020, subsistence farms or very small farms employed seventy-five percent of farm managers in the EU who were 65 years of age or older (Figure 4).

According to our analysis, there are wide differences in the percentage of younger and older farmers in EU member states. EU countries were categorized into four groups based on the age of farmers. Countries with a high percentage of young farm owners and a low percentage of elderly farmers were included in the first group (France, Poland, Luxembourg, Finland, Austria). The second group consisted of nations having a significant portion of older farmers in addition to a fair number of young farmers (Spain, Slovakia, Germany, Bulgaria, Ireland, Hungary). The third group comprised countries where the percentage of younger farmers was low (Czech Republic, Italy, Belgium). Lastly, the fourth category consisted of countries with a very low percentage of young farmers and a relatively large percentage of older farmers (Portugal).

The agricultural sector's economic performance [26–29] was evaluated using net value added at factor cost, which is gross value added that is modified to account for output taxes, subsidies, and fixed capital expenditure. Because it represents the compensation available for all components of production, it is often referred to as "factor income" (land, capital, and labor).

It is possible to state factor income per full-time labor equivalent. It is considered a statistic of partial labor productivity since it measures the net value added by the equivalent of each full-time worker in the agriculture industry. This productivity measure, known as Indicator A, is given as an index and is computed in real terms following the deductibility of inflation.

In order to comprehend the evolution of this measure of agricultural revenue, one must first comprehend the evolution of agricultural labor that agricultural revenue is theoretically distributed over. Given the large quantity of seasonal, part-time, and unsalaried labor input in agriculture, the annual work unit is the most appropriate unit of measurement for describing the actual amount of work performed in farming operations. The amount of work completed is expressed in full-time work equivalent with this measure.

In 2022, the EU's workforce input for agriculture amounted to 7 million full-time workers. Non-salaried labor made up the majority of the entire labor input in agriculture;

in 2022, there were around 5 million full-time workers in this category. In 2022, there were about 2 million full-time workers who were salaried.

Most EU members utilized less labor overall in agriculture in 2022 than they did in the year before, with Bulgaria (7.0%), Hungary (9.2%), and Spain (6.0%) seeing especially sharp declines. However, in several nations, including Slovakia (7.0%), Estonia (2.0%), Latvia (2.4%), and Lithuania (2.6%), the amount of labor utilized in 2022 was anticipated to have been larger than in 2021.

A few EU members, most notably Slovakia, Cyprus, and Lithuania, employed more paid farm labor in 2022 than in 2021 (Figure 5). This trend was partly caused by higher employment demands at seasonal peaks. This was frequently in opposition to the general decrease in the quantity of labor employed in agriculture.

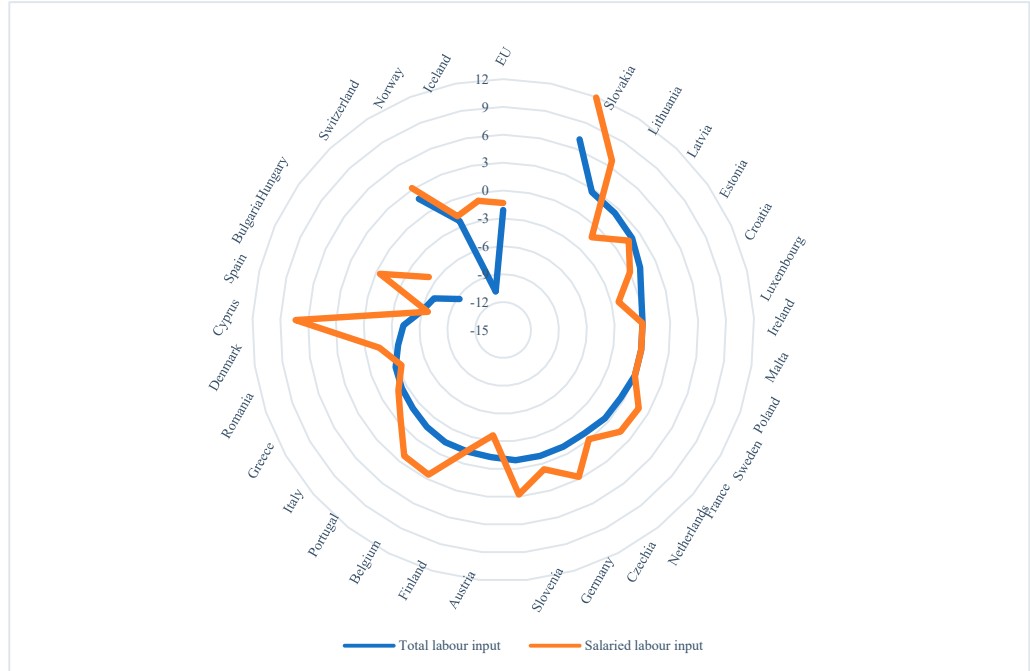

**Figure 5.** Labor input for agriculture (annual rate of change in percent, 2021–2022, authors' analysis based on Eurostat data [24]).

The data show (Figure 6) that the countries with the worst decline rates were Malta (9.0%), Portugal (10.5%), and Romania (22%). The EU saw increases in the agricultural income of other member states, such as Germany (57.8%), Poland (23.6%), France (11.5%), Italy (9.0%), Luxembourg (32%), Estonia (30.0%), Sweden (26.0%), Austria (25.0%), Ireland (16.7%), and Slovenia (15.0%).

A decoupling of economic growth from resource usage and its environmental effect is necessary for an economy to become more sustainable [30]. Prices need to represent the true costs of resource consumption in order to properly assess this sustainability [31,32]. However, farming operations seldom have a price attached to their social and environmental outcomes. In fact, neither the so-called "green accounts" nor the potential green efficiency indicators for agriculture are close to being finished. The same is true for total factor productivity statistics in agriculture, which compare agricultural output to a composite measure of input from labor, capital, land, and intermediate consumption.

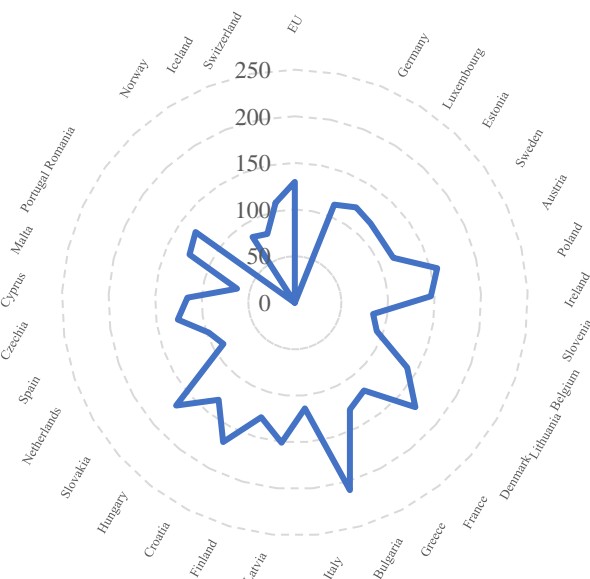

**Figure 6.** Agriculture's income per work unit annually (2015 = 100, 2021–2022, authors' analysis based on Eurostat data [24]).

The agricultural sector's output volume in the EU had a growing trend from 2007 to 2021; however, it began to decline in 2022 (Figure 7). Using the EAA approach, we can use changes in the volumes of outputs produced and in the products and services consumed as inputs in the production process to make inferences about the resource performance of agriculture. These quantities result from breaking down the values into their price and volume components. These implied volumes have no unit of measurement—kilograms or tons—and are not quantities. It is crucial to remember that they are called "volumes" because they record changes in amount, quality, and composition in addition to quantity changes. Volumes may be used for various productivity and performance metrics since, as indexes, they offer a broad picture of the changes in the quantities of inputs and outputs [33,34].

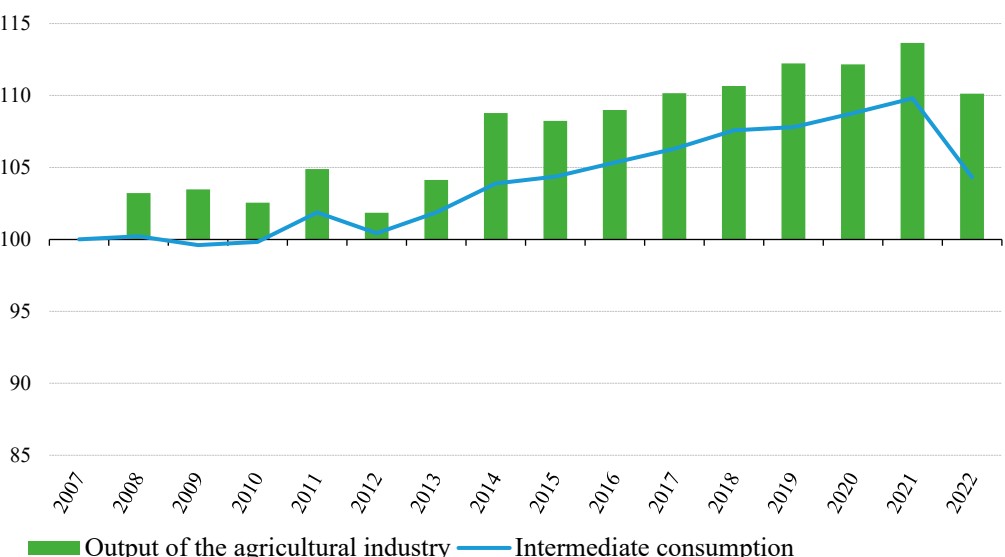

**Figure 7.** Volume indicators for intermediate consumption and agricultural output (2007 = 100, EU, 2007–2022, authors' analysis based on Eurostat data [25]).

The decline in the number of youths employed in agriculture creates particular problems for the renewal of generations and raises concerns about the knowledge and skills that will be lost as older, more seasoned workers retire. The literature's conclusions led to the

recommendation that training programs for farmers should concentrate on their preferred learning styles as well as particular needs. For example, Sotte [35] argued that young farmers have particular challenges that call for advice and information specific to their situation. This is in line with the assertion made by Klair et al. [36] that the advancement of the Common Agricultural Policy is closely correlated with farmers' information needs. The EU legislative framework for rural development 2014–2020 states that knowledge transfer and information acts should be customized to the needs of rural actors rather than being provided as standard training.

It has been observed that, to satisfy farmers' needs, agricultural training programs must be adjusted [37–39]. "It is evident that paying greater attention to what the farmer regards as his/her requirements would be desirable if extension services are to be more effective", argued Brent and Adams [37]. The demands of farmers must also be considered in other contexts, such as in the foundation of agricultural research [40] or the effective use of field margin measures, according to Mante and Gerowitt [41].

Additionally, it is essential that young people obtain adequate training on emerging issues in the EU's agrifood system. These might involve the following:

(a) Climate change: How rising temperatures, droughts, and other effects of climate change could impact productivity differently than they did in the past.
(b) The energy industry is subject to changing hazards, including fluctuations in diesel and gasoline prices that impact transportation, machine operations, and other expenses, as well as competition from biodiesel.
(c) Evolving customer preferences about organic food, food safety, and traceability, among other issues.

To meet the needs of young farmers, training programs and other knowledge transfer methods must be modified. Determining the intended recipient of the knowledge is also essential since different types of farmers need different kinds of information and instructional strategies [42–45]. Certain farmers may not benefit from exchange programs or other interactive instructional initiatives. The farmer's skills and context-specific factors, such as the institutional framework in which the farmer works, resource availability, and recognized opportunities and risks, will determine this appropriateness [46].

### 3.2. Instruments, Tools, and Model Development
Income Risks and Management Instruments

Hazards related to income are influenced by a variety of elements, including the environment, as well as the patterns of pricing and production risks (farm management, entrepreneurship). Many variables, such as structural change, climate, and financial risk, have an impact on income hazards; however, they vary depending on the location. The agricultural production structure, or sector mix, and the level of heterogeneity within each sector vary between member states.

There is a great deal of heterogeneity in fluctuations in incomes both within and between the farming sectors of member states, as well as in incomes of individual farms over time. This variance is brought about by both farmer- and farm-related variables, such as farm management skills (knowledge, business) and farm features, as well as external forces such as market movements, production risk, environment, pathological, and phytosanitary issues (size, location). Income variability at the member state level is determined by the mix of agricultural output sectors and the degree of variability within each sector [47].

EU member states fund various agricultural risk management initiatives [48], which aid in stabilizing their agricultural sectors. Farmers can use a variety of risk management tools, including insurance, mutual funds, savings accounts, ad hoc basis payments, and budgetary controls (Figure 8). Except for Greece and Cyprus, where insurance is required by law, the majority of traditional insurance programs in the EU are private, mostly consisting of single-risk and combination insurance as well as yield insurance. Each nation's degree of government agricultural growth is primarily determined by two key

factors: the requirements (risk level) of every nation and the financial assistance provided to the insurance systems by every member state.

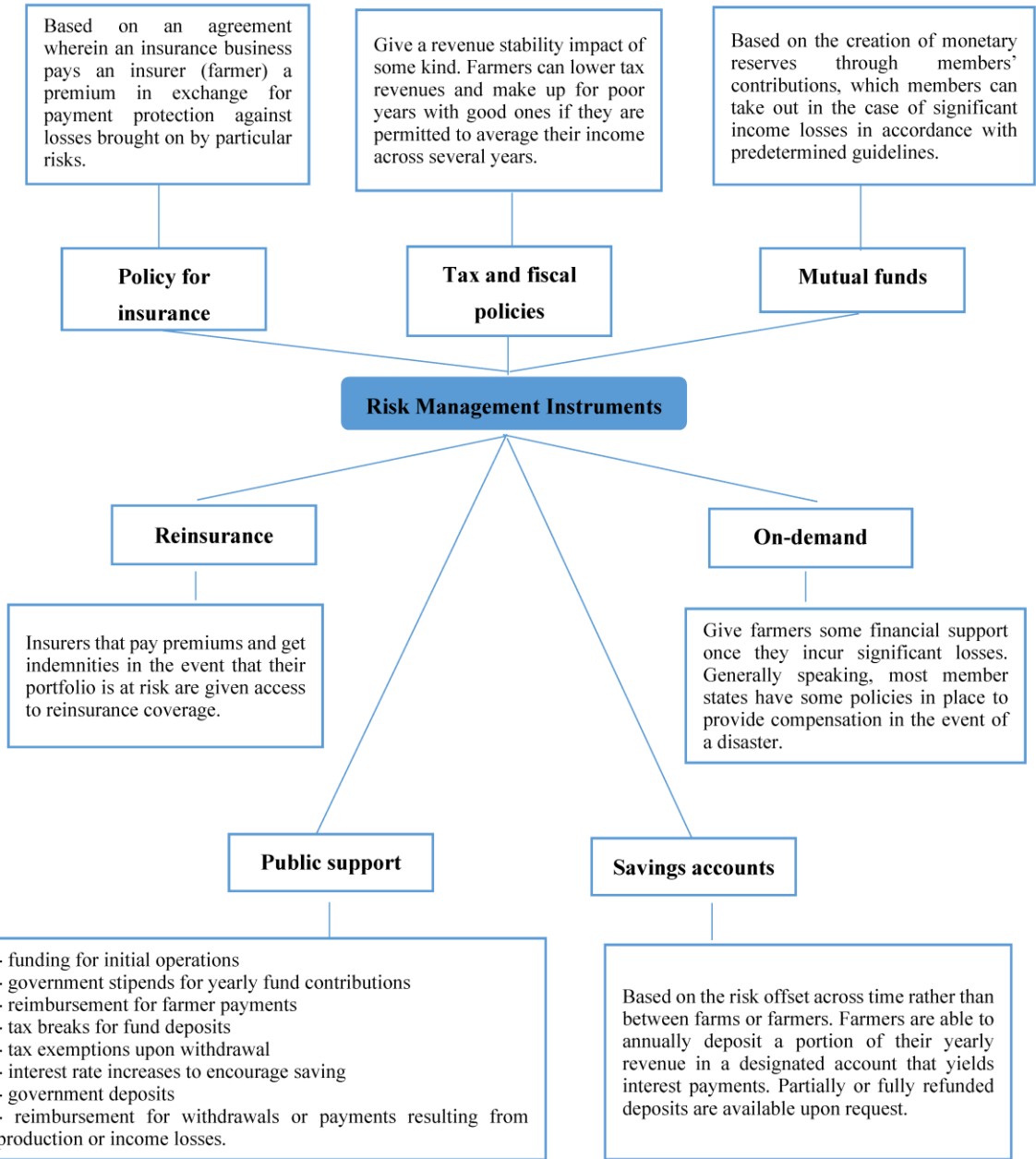

**Figure 8.** Tools for risk management; authors interpretation according to the EU support policy in the agricultural sector (authors' own development).

State aid is used by all EU member states as a crisis management tool. There is a glaring underutilization of ex ante (risk) management measures in member states, since the majority of them solely base their public help on state aid (ex post measures devoted to crisis management). As might be predicted, the states with the highest levels of agricultural risk rank highest in terms of total public spending. In terms of total public aid, Spain is the largest, followed by Greece, Italy, France, the United Kingdom, Poland, and Germany. Cyprus, Slovenia, and Greece appear to have the most public support for risk management when measured in terms of relative value to the system, that is, in relation to the quantity of agricultural output.

Regardless of insurance policies, public assistance or compensation for agricultural losses are provided on an as-needed basis in the majority of nations. In the event that

insurance is not subsidized, compensation plans or catastrophe bonds—which are sometimes partially funded by agricultural stakeholders—are frequently used to give assistance (on a voluntary or compulsory basis). Generally speaking, the livestock industry has less developed insurance plans than the agricultural industry, with a primary focus on accidents and non-epidemic illnesses. Sanitary support programs are essential for managing livestock risk; public help is provided for big crises or illnesses with significant externalities. Because reinsuring systemic risks like drought is expensive, it poses a challenge to agricultural insurance plans. Certain nations (Spain, Italy, Portugal) provide public reinsurance of some type.

## 4. Model for Agricultural Risk Management

Farmers face numerous risks, which can be summed up as follows:

- Market risk stemming from changes in market conditions, such as fluctuating pricing and increased volatility.
- Production risk resulting from weather unpredictability, animal sickness, and plant pests.

Hazards are divided into two groups based on different damage thresholds: "normal" or non-severe losses (less than 30% of yield or revenue/income) and severe losses (more than 30% of yield or revenue/income). As a result, insurance companies consider revenue or income insurances to be less desirable and riskier.

The foundation of the agricultural risk management layering approach is the idea that various risk levels, or layers, ought to be handled by designated players using particular tools and funding sources. The most crucial requirement for a layer-based risk management system is to guarantee the system's overall balance and consistency by making sure that each layer's instruments and the actors involved in them are compatible with one another. It also needs to be consistent with other policy measures to prevent discouraging the relevant actors from managing the corresponding layer (Figure 9).

The diverse nature of risk profiles among member states necessitates a generic portfolio of instruments for ex-ante risk management to effectively cover agricultural hazards. There is no single risk management strategy that can accommodate every farmer's requirements and circumstances in every EU member state. To guard against typical (shallow) and unusual (deep) income losses, an ex-ante risk management portfolio can consist of the following:

- Conventional non-financial on-farm risk management techniques with on-farm protection against shallow losses, employing precautionary savings and/or income tax smoothing possibilities (e.g., agronomic practices such as diversification).
- Long-term contracting techniques assist in managing risk throughout the food chain.

Mutual funds, like the income stabilization tool or creative insurance contracts, might provide sufficient support for the sustainability of such contracting procedures, which are susceptible to severe market occurrences (extremely high or low market prices).

Mutual funds, contractual pricing agreements for more systemic risk and insurance against unique hazards, are a way to shift off-farm risk. Margin, revenue, and yield protection may be included in mutual fund and insurance coverage (further study and assessment would be needed to explore the potential opportunities of revenue and margin insurance).

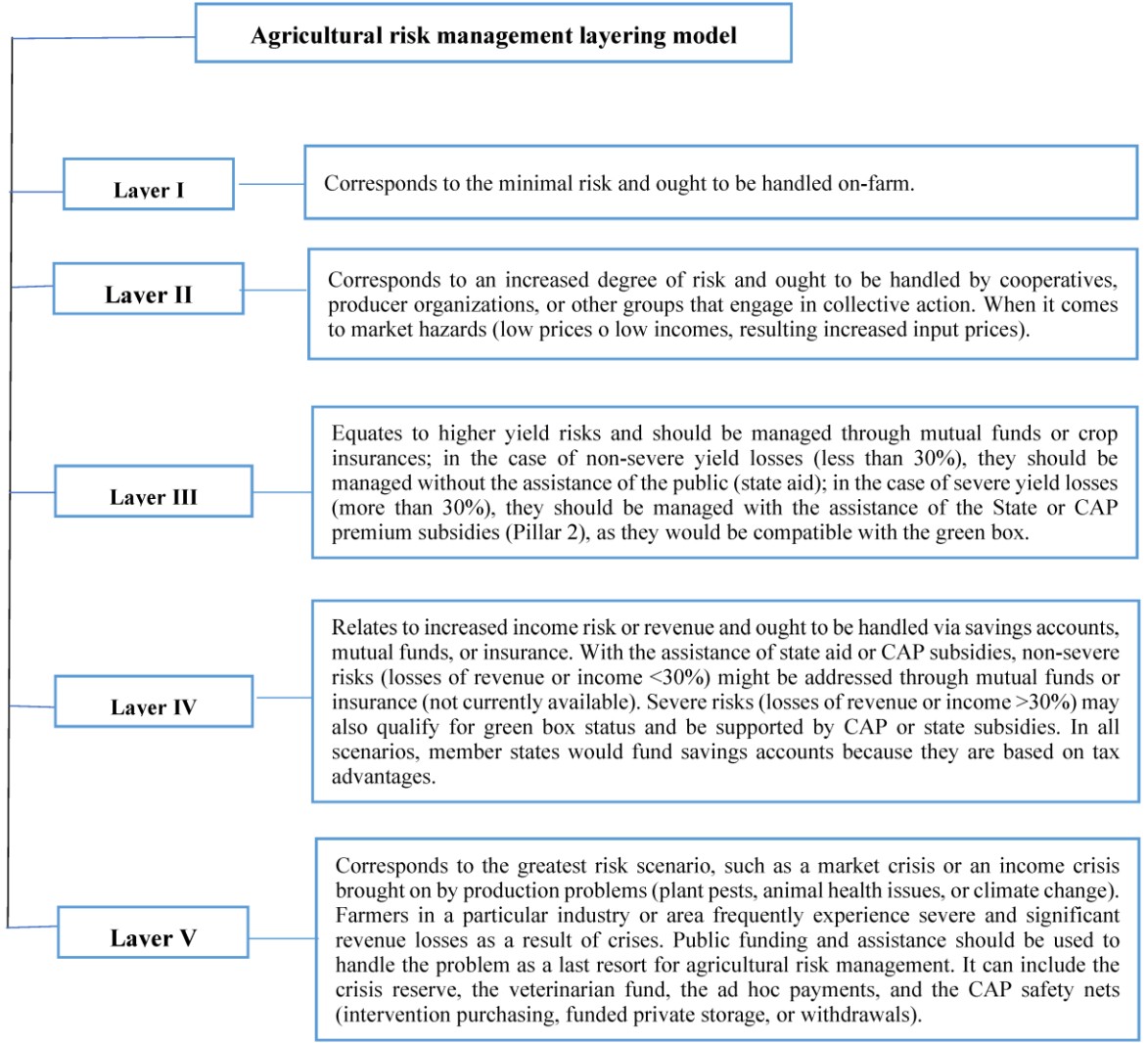

**Figure 9.** Agricultural risk management layering model (authors own development based on the European Commission management layering model in agriculture).

### 4.1. Income Stabilization Tool

The alternative method offered by the Rural Development Regulation to mitigate income hazards is the income stabilization tool (IST). The IST is a mutual fund that was included in the post-2013 reform and serves as an income loss-compensated vehicle. Farmers contribute to the fund to create a financial reserve that may be utilized to reimburse them for revenue losses, regardless of the reason.

When the effective loss is greater than 30%, relative to the average yearly income of the preceding three years or the "Olympic" average of the preceding five years, the IST only makes up less than 70% of the lost income.

Up to 65% of the qualifying costs—which include the administrative expenses associated with establishing the fund and the cash compensations given to farmers—may be funded by the EU under Article 39. The fact that funds may only be used in cases of income losses of more than 30% is one of the primary barriers to the adoption of the IST. Moreover, the inability to tailor IST programs to certain industries hinders their ability to be programmed, even if the risks and likelihood of receiving payouts vary greatly throughout agricultural sectors. Additionally, the original capital stock cannot have any public support.

The IST is seen as a difficult instrument, impeded, among others, by the widespread lack of expertise and information on how to develop it. Governments and farmer organizations often have questions about organizing, starting, and running mutual funds. Another

issue is what should happen when money needs to be paid in one of the initial years when the capital stock is still relatively modest. Furthermore, the IST's financial requirements might be quite unpredictable and stringent.

Market price indices for inputs and outputs, weighted by the disclosed account structure ratios during the reference period, might be used to determine the actual loss. To get closer to the reality of farmers, meanwhile, would mean a huge administrative load and the need to calculate market price indices at a smaller regional scale. The Commission suggested many changes to the Omnibus plan to encourage the adoption of the IST. With these enhancements, member states will have the option to tailor the IST to a particular industry, lower the compensation threshold from 30% to 20%, and explore the idea of allowing public contributions to the initial capital stock.

Measuring the expected/guaranteed and actual revenue/income is a major challenge for the income stabilization tool (IST), whether it is used with mutual funds or revenue/income insurance.

*4.2. Support Tools to Help Young Farmers*

The decline in the number of young individuals employed in agriculture creates particular difficulties for the renewal of generations and gives rise to concerns over the loss of important expertise and knowledge when older, more seasoned workers retire. Due to their high startup costs, difficulties securing funding, and low turnover in their first few years of business, young farmers face particular challenges. This might discourage aspiring young farmers from entering the agriculture industry, combined with longer generational regeneration and restricted access to land.

According to Gonzalez and Benito [49], McDonald et al. [50], Vesala [51], and other scholars, young farmers approach farming with a considerably more entrepreneurial mindset and are particularly driven to establish new markets [52]. Additional studies demonstrate the impact of young farmers on the uptake of sustainable agricultural practices [53–55] and their contribution to resource and land efficiency [56]. With more positive assessments resulting in more effective solutions, the "young farmers dilemma" may instead be seen as the "young farmers challenge" [57].

In addition to increasing EU agriculture's competitiveness going forward, fostering the next generation of European farmers also contributes to securing the continent's food supply for years to come. Particular EU programs aimed at supporting the formation of new farmers have helped policymakers recognize the need to address the issue of young farmers. Regarding this, the 2014–2020 CAP amendment and the 2007–2013 CAP reform both included provisions targeted at young farmers and new entrants. These policies include the Young Farmers Payment (Pillar 1) and the Start-up Aid for Young Farmers (Pillar 2).

Start-up aid for young farmers, unlike the Young Farmer Payment within Pillar 1, has been available in the EU since the 1980s. The policy instrument is part of the Rural Development Policy, as required by Regulation (EU) No. 1305/2013. Support for young farmers is included in Priority 2 (farm viability and competitiveness), along with an emphasis on generational renewal and Focus Area 2B (facilitating the entry of properly skilled farmers into the agricultural industry). To accomplish the objectives stated in this target area, programs for rural development use a range of tactics. Members may encourage business start-up assistance for aspiring farmers under this provision.

Pillar 1 Young Farmer Payment financial aid was implemented as part of the 2013 CAP reform. The Young Farmer Payment (YFP) is a mandatory program that all EU member states need to adopt under Regulation (EU) No. 1307/2013, which mandates that they devote up to 2% of their direct payments budget to the YFP. The majority of member states have chosen to pay 25% of the average direct payment per hectare and have set the maximum number of payment entitlements or hectares at 90. Some member states reported a maximum of 2% of the direct payment envelope, while others reported less than 1%.

Measure 112 (Common Agricultural Policy, Pillar 2) of the 2007–2013 CAP emphasized the growth of farms run by farmers under the age of 40 [9]. The most important support was provided under CAP's Measure 6, "Business start-up help for young farmers", from 2014 to 2020. The European Network for Rural Development [58] released figures showing that, between 2007 and 2012, Measure 112 supported over 125,000 young farmers in the EU, with France accounting for over 26,400 of those cases [59].

Originally intended to encourage family farm succession, early retirement programs were implemented by the 1992 CAP modifications under rule 2079/92. These programs assisted in older farmers' retirement, mostly in southern Europe [60]. Afterward, Measure 113, the "Early retirement plan" (Common Agricultural Policy, Pillar II), was put into place during the 2007–2013 CAP to incentivize older farmers to retire early by offering a pension of up to EUR 15,000 annually for a maximum of ten years [61].

As part of the 2013 CAP reform, Measure 113 was phased out due to its inability to foster intergenerational transfer [25]. The young farmer payment had a maximum value of EUR 5400, with a baseline payment of EUR 240 per hectare in lowland areas. According to Andersons [62], the assistance that the present CAP amendments offer to young farmers is mostly pecuniary and has few long-term effects.

Between 2007 and 2020, the EU committed EUR 9 billion mostly to support young farmers to increase agricultural holdings' competitiveness and promote generational renewal. The total amount of public investment, including co-financing from member states to set up Pillar 2 initiatives, was EUR 18.3 billion. The European Union gave start-up financing to almost 200,000 young farmers between 2007 and 2013. Between 2014 and 2020, the European Union committed almost 70% of its money to the Pillar 2 (EAFRD, European Agricultural Fund for rural development) measure to recruit young farmers [63]. The remaining 30% of the funds was assigned to the Pillar 1 (EAGF—European Agricultural Guarantee Fund) program, which provides direct payments to young farmers. Additional financing alternatives for farmers, such as bank guarantees and preferential loans, are permitted by the EAFRD rule [64–66]. DG AGRI's (Directorate-General for Agriculture and Rural Development) young farmers' project [9,67,68], in partnership with the European Investment Bank (EIB) Group, aims to combine all available EAFRD support with the EIB Group's financial resources and expertise. The principal components of the initiative consist of:

- A new EIB lending package for farmers, facilitated by specialized intermediary banks, featuring a special envelope component for young farmers;
- The ongoing utilization of EAFRD grants for young farmers and start-ups, which can also be applied to interest rate subsidies or technical support;
- EIB advisory support to EAFRD managing authorities, either bilaterally or through fi-compass.

By utilizing the European Investment Fund (EIF) fund management capabilities of EAFRD-backed financial instruments, managing authorities may also be able to make use of extra funding and expertise from specialized financial intermediaries.

Together with the European Commission and the EIB, DG AGRI provides the Fi-compass advisory platform. Together, the three organizations hope to increase the public awareness of these combined funding and advisory options and provide support to a select group of managing authorities and banking intermediaries so that they can test them out.

The European Commission conducted an evaluation in 2021 of the effects of the CAP on jobs in rural areas, local development, and generational renewal. Examining whether the CAP's tools are adequate and being used wisely to promote job creation, local development, and generational renewal in rural areas was the main objective of this evaluation and took into account the most significant generational renewal programs, which are as follows: (i) direct financial support to young farmers; (ii) investment help; and (iii) company start-up aid (CAP generational-renewal measures).

Several challenges and limitations faced by the review included a lack of comprehensive and reliable data, a short observation period, and limited geographic coverage of several studies. According to the analysis, generational renewal measures of the Common Agricultural Policy (CAP) have a generally good effect on the number of young farmers, albeit a slight one in some regions, especially those without basic infrastructure and services. This beneficial effect is influenced by several other factors, including sociocultural and wider economic incentives and disincentives for people to farm and live in rural areas [69,70].

Recent studies show that the gender of married spouses affects farm profitability (with male couples achieving the best profitability) [71], although female farm succession is still significantly less prevalent than male succession [72].

Programs for the generational renewal of the Common Agricultural Policy (CAP) therefore cannot be evaluated in a vacuum, divorced from other EU policies, the larger socioeconomic context of rural areas, and state laws and regulations. The CAP's generational renewal programs improve farm business performance, resilience, and the secure handoff of farms to future generations (Figure 10). The policies are more likely to encourage the socioeconomic stability of farm companies once young farmers have established their operations, as opposed to fostering farm succession. Moreover, off-family farm transfers are not appropriate for CAP generational-renewal approaches.

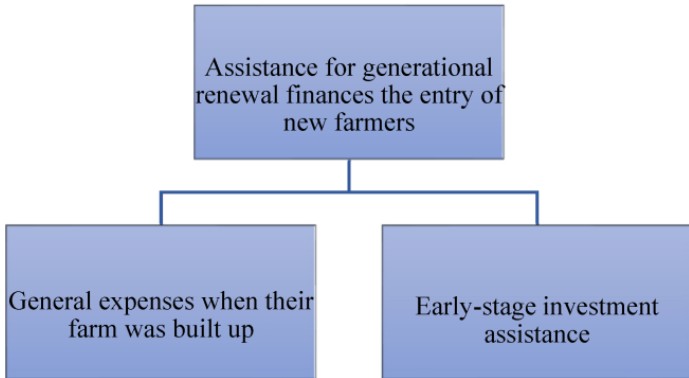

**Figure 10.** Promotion of generational rejuvenation through CAP (authors' own development).

To make land and money more accessible, national legal, social, and economic policies may need to be more aligned with the goals of the CAP generational renewal. Support for rural development, together with other EU initiatives like the cohesion and regional grants, which promote added value, improved services, improved infrastructure, and rural economic diversification, can enhance the overall economic environment, particularly in isolated regions. The strongest evidence of the long-lasting and beneficial effects of CAP support can be found in the following areas: financing and investment aid for business start-ups; guidance and training; incentives for cooperative institutional arrangements and/or fiscal arrangements that facilitate intergenerational transfer.

The implementation and efficacy of CAP generational-renewal initiatives might be enhanced by member states through the development of integrated approaches that incorporate various CAP and non-CAP instruments, institutions, and wider legislative and budgetary requirements. There are clear distinctions between the Pillar 2 setup measure and the Pillar 1 payout to young farmers (Figure 11).

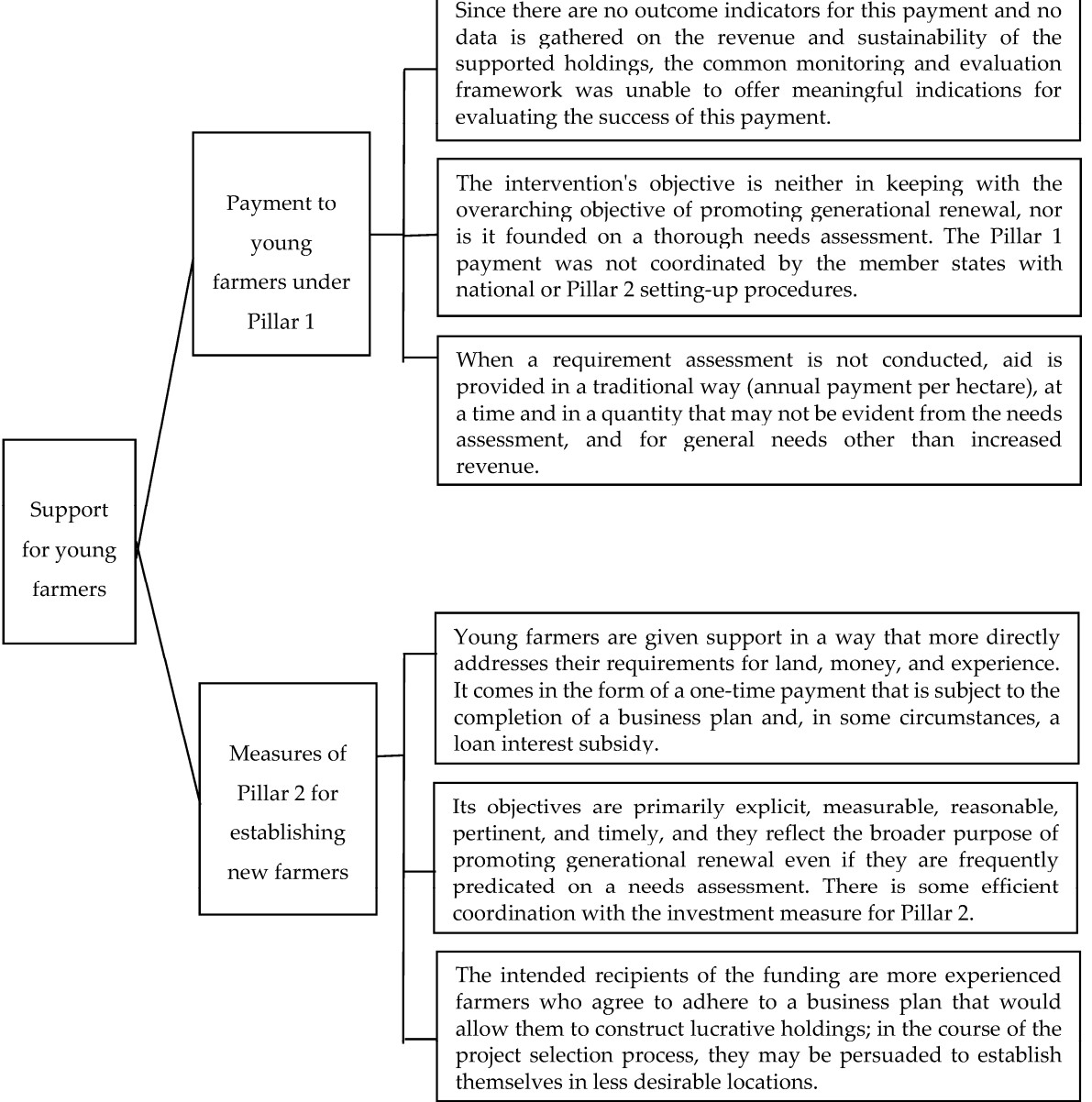

**Figure 11.** Disparities between the Pillar 1 payment to young farmers and the Pillar 2 payout (authors' own development).

### 4.3. Support for Agricultural Risk Management

The European Union possesses a versatile legislative structure to facilitate risk management tools (Figure 12), enabling member states to effectively address a wide range of agricultural hazards. The CAP and the regulations governing state support to the agriculture sector define this framework.

Pillar 1 only considers fruit, vegetable, and wine production when it comes to the potential to enhance risk management tools; Pillar 2, or state aid, takes into account other sectors. There is a significant distinction between state aid and support based on Pillar 2. In the initial instance, support must be categorized as green box policy in accordance with WTO regulations; however, amber box support is permitted under the State Aid Framework. When subsidies are "decoupled" from current production levels, they are referred to as "green box" schemes because they contain set payments to producers for environmental programs. Subsidies known as "amber boxes" have the power to stifle

international commerce by driving down the price of a nation's goods relative to those of other nations.

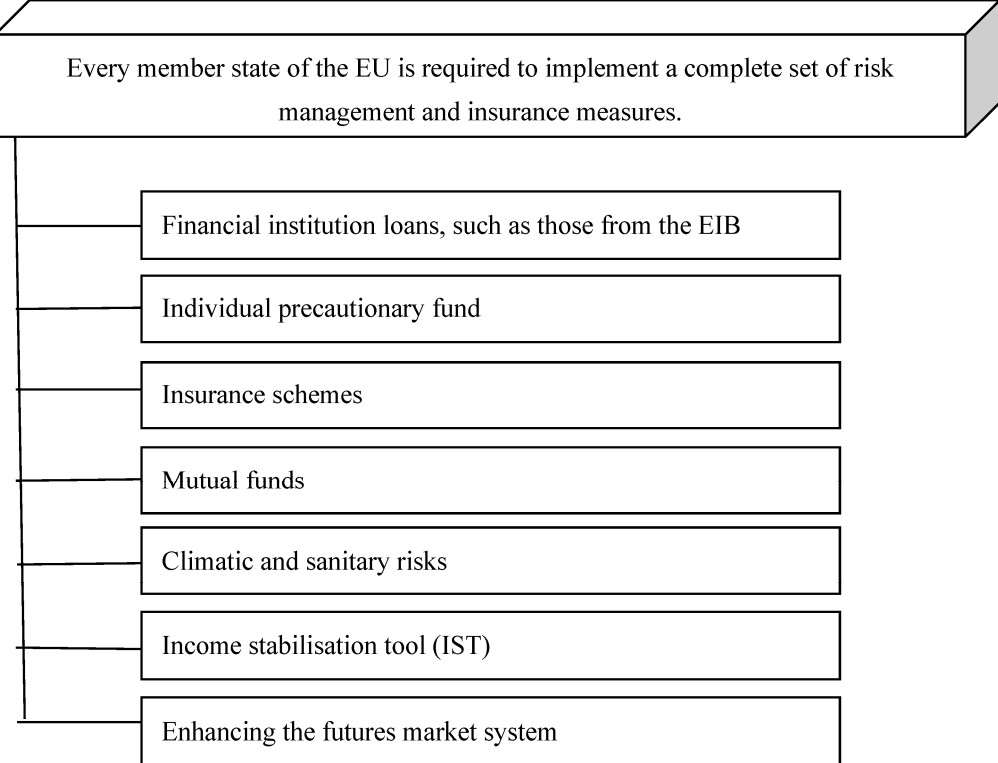

**Figure 12.** Risk control and protective measures (authors' own development).

The longest measure is the "crop, animal and plant insurance premium". Mutual funds are the sole option available in the EU to offer coverage against market risks (Figure 13). This is a serious constraint since it eliminates other ways to provide market risk coverage, such as income/revenue insurance or savings accounts. Because of the tight budget and inconsistent execution, the inclusion of risk management tools in Pillar 2 presents some implementation challenges. With tools included in both Pillars, a complete and consistent risk management policy is not established.

As part of the support tools, it is also necessary to monitor various economic features of farms using farm level data, relevant proxy indicators, and farm economics research. Here are several examples:

a.  Agricultural household simulation models (e.g., Takeshima et al., 2013) [73] that can be used to ex-ante analyze the prospective viability of new technologies given the farm's current technology levels and resource restrictions.
b.  Production technology returns to scale and how they are affected by various aspects (e.g., Takeshima 2017) [74]; if the technology utilized by young farms has sufficient returns-to-scale, it can incentivize young farmers to solve farm expansion constraints.
c.  A measure of flexibility (also associated with economies of scope) in terms of alternating between crops, modes of production, or between farming and other enterprises (Takeshima et al., 2022) [75]. The estimation of the production function yields the flexibility indicator. The resilience indicator, which measures the potential for agricultural diversification to lessen the negative consequences of risks, and the flexibility indicator are closely related.
d.  The technical effectiveness of agricultural businesses, their susceptibility to price shocks, and the variables reducing that susceptibility (using Central Asian examples, such as Kazakhstan and Uzbekistan, which may be applicable to Eastern European nations; see, for example, Takeshima et al., 2023) [76].

e.      A model to evaluate and track managerial ability (e.g., Nuthall 2009) [77] highlights that a farmer's management style, early life experiences, and exposure to events all play a major role in determining ability. Apart from the various limitations stated by the writers, a significant hindrance for young people is the lack of farming experience and administrative skills.

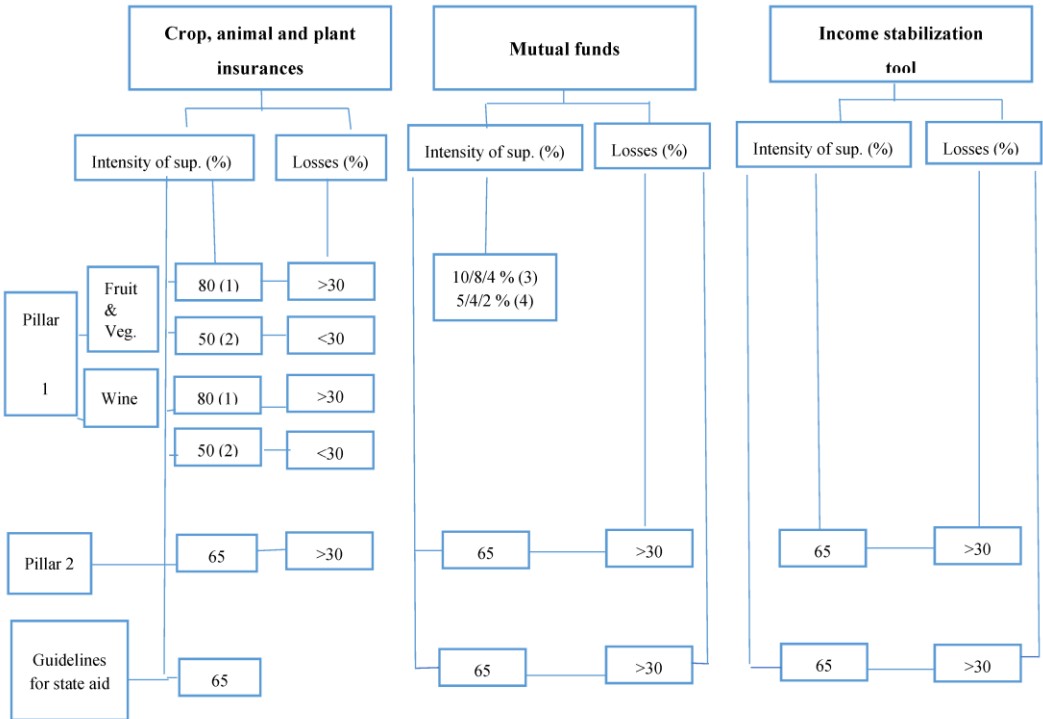

(1) Unfavourable weather conditions that are linked to natural disasters

(2) Additional damages brought on by weather conditions, illnesses of animals or plants, and plant infestations

(3): Percentage of the producer organization's contribution to the mutual fund during the fund's first, second, and third years of operation in the event that member state joins the EU after 2003

(4) A part of the producer organization's contribution to the mutual fund during the its first, second, and third years of existence for the remainder of member states.

**Figure 13.** Assistance with risk management during 2014–2020 (authors' own development based on European Commission, 2021 [2]).

## 5. Conclusions

In order to secure farming's long-term sustainability, profitability, and productivity, young farmers are prepared to embrace innovation, smart agriculture, and science-based research. The CAP's policies need to support intergenerational collaboration, offer unambiguous exit routes, and encourage generational renewal.

For the sustainability of the agricultural industry in Europe, there must be an EU-wide strategy for youth access to agricultural land. The results of the distribution of young farm holders among the different size categories of holdings illustrate that the largest cohort of young farmers in Europe operate farms belonging to the smallest-sized category. Nowadays, the biggest barrier for young people who want to work in agriculture is land availability, which is made worse by the fact that young people in the EU have limited access to funding. Succession needs to be encouraged and made simpler in order to combat the aging trend in the agricultural population and ensure future food supply in Europe and beyond. New forms of cooperation between agricultural generations might be fostered with the support of partnerships, land mobility services, long-term leasing and other long-term agreements, share-farming, and farm-to-farm relationships.

Young farmers often require financial assistance in order to obtain loans for investments and other inputs. They require more access to low-interest short- and long-term financing.

- New financing models that incorporate companies in the food and agricultural industries and offer creative financial solutions to help farmers during challenging times.
- Increased cooperation with the European Investment Bank (EIB) and the European Investment Fund (EIF) to support the development of financial solutions aimed at young farmers across the EU.
- Public assurances, ensuring they will not have to fear missing out on CAP growth prospects.
- Commercial banks are asked to offer them better financing rates and services by means of potential state incentives.

Additionally, member states need to support young farmers more vigorously and to promote land mobility and generational renewal in the agriculture sector, especially with regard to taxation, land management, laws governing land inheritance, sales of farms, and rental expenses. Member states have to create a mandatory program to assist newly arrived young entrants in getting started, with a focus on youth who do not come from farming families. It is necessary to create legislation that will facilitate the implementation of national land mobility and succession planning initiatives. Managing authorities are required to regulate these services.

In the agriculture sector, young farmers' comprehension of the nature of different hazards is improving, as is their access to education and information services. Most of the volatility in their line of work can be traced back to a pattern of extended low price periods and significant revenue fluctuations, which have built up uncertainty. Young farmers are more financially exposed, making them especially vulnerable under these circumstances. Therefore, it is necessary to provide young farmers with strong decision-making tools and reliable, high-quality market information so that they can effectively use risk management tools and control their own risks.

Young farmers need access to strong decision-making tools and reliable, high-quality information in order to effectively employ risk management tools and self-manage their risks. It has been determined that the following risk management techniques for CAP beyond 2020 will assist young farmers in preserving and enhancing farm profitability.

Within the EU, the risk varies greatly between member states and also depends on the size and type of farm. Since the current risk management techniques are typically insufficient to mitigate an income drop, member states nevertheless provide ex post aid. A uniform regulatory framework, appropriate control mechanisms, public reinsurance, partial subsidization of national systems, and the facilitation or subsidization of database composition at the farm level are some measures that might be implemented to support national systems.

To develop instruments that reduce risks in EU agriculture, three issues must be addressed: the ability of the stakeholders to develop, implement, and oversee risk management instruments; the new instruments' possible impact; and the integration of the new instruments within the larger policy framework.

The integration of risk management tools into national and EU policy frameworks, as well as the relationships between these tools and other interventions (such as direct payment and ex post instruments), must be carefully considered in future policies. It is also important to take into account how well they mesh with global obligations. It is crucial to assess the possible effects of the widespread use of risk management tools and to quantify the predicted changes in farm revenue and the volatility of agricultural commodity prices. The framework of a more comprehensive risk management plan should also take into account the environmental influence or unexpected repercussions. The creation of ex ante and ex post assessments and studies of the risk management policy choices made by the member states should be encouraged in order to ensure that policy decisions are grounded in evidence.

Knowledge transfer and advisory services in areas including technical, economic, environmental, and social awareness regarding succession concerns should be available to farmers. To assist young farmers in getting ready, free and open advisory services in a variety of areas (such as land management, inheritance law, business support, marketing counsel, and other financial advice) may be made accessible.

Frameworks that incentivize farmers to share their experiences are critical to the effective and efficient transmission of knowledge. Farmers need to have continuous training and education available to them. The modernization of vocational training is required in rural areas. In order for customers to comprehend EU legislation and the quality of food production, farmers and consumers must communicate. Thus, the following criteria apply:

- It is crucial to communicate and be transparent so that consumers are aware of farming practices, the methods used to produce food, and the ways in which the CAP benefits the larger society.
- A structured system of information exchange must be made available for both individuals and groups of farmers from various areas, industries, and generations.
- Farmers must be offered continuing professional development through training and upskilling; in the event that there is no advisory service, a reasonably priced one should be set up to maximize the production systems. Either the state or the private sector might own this.
- Young farmers should be able to use vouchers to obtain these services.
- Advice and information sharing should not be limited to money matters, qualifying standards for investments, or startup funding.

The ability of young Europeans to invest and create is critical to the future of rural communities. In order to prevent the farming population from aging and to ensure farm success, financial support must be provided through instruments that are specifically aimed at young farmers. Examples of these instruments include guaranteed loans and preferential access to credit. The ultimate goal is to have a multifunctional agriculture sector that supports individual farmers, farm families, and cooperatives.

**Author Contributions:** Conceptualization, A.F. and A.T.; methodology, A.F. and A.T.; formal analysis, A.F. and A.T.; resources, A.F. and A.T.; writing—original draft preparation, A.F., A.T. and K.M.; writing—review and editing, A.F., K.M. and A.T.; visualization, A.F.; supervision, A.F. and A.T. All authors have read and agreed to the published version of the manuscript.

**Funding:** This research received no external funding.

**Institutional Review Board Statement:** Not applicable.

**Informed Consent Statement:** Not applicable.

**Data Availability Statement:** Data are contained within the article.

**Conflicts of Interest:** The authors declare no conflicts of interest.

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
