# Peer review of "A Sustainable Risk Management Model and Instruments for Young Farmers in EU Agriculture"

_sustainability, doi:10.3390/su16010283_

Round 1

Reviewer 1 Report

Comments and Suggestions for Authors

Comments on the Quality of English Language

Minor revision

Reviewer 2 Report

Comments and Suggestions for Authors

Thank you for the opportunity to review the manuscript. Please find my comments in the attached referee report.

Comments on the Quality of English Language

Reviewer 3 Report

Comments and Suggestions for Authors

The article discusses a very original topic. The authors therefore found an interesting and valuable research gap. They actually justified it and supported it with good literature studies. Agriculture is crucial for the quality of life of societies. It also determines the future of nutrition. Preventing young people from leaving the sector is therefore extremely important theoretically and practically. The risk assessment methodology proposed by the authors is very good, transparent and correct. The graphic material allows you to better understand the presented content and is carefully prepared. The literature is extensive and properly used. However, the authors could have articulated the purpose more precisely in the abstract and deepened the discussion to enhance the scientific value of the text.

Round 2

Reviewer 2 Report

Comments and Suggestions for Authors

Thank you for addressing my comments. I think the authors addressed them well and I have no further comments.